

# Growth of hybrid open access, 2009–2016

Bo-Christer Björk

Information Systems Science, Hanken School of Economics, Helsinki, Finland

## ABSTRACT

Hybrid Open Access is an intermediate form of OA, where authors pay scholarly publishers to make articles freely accessible within journals, in which reading the content otherwise requires a subscription or pay-per-view. Major scholarly publishers have in recent years started providing the hybrid option for the vast majority of their journals. Since the uptake usually has been low per journal and scattered over thousands of journals, it has been very difficult to obtain an overview of how common hybrid articles are. This study, using the results of earlier studies as well as a variety of methods, measures the evolution of hybrid OA over time. The number of journals offering the hybrid option has increased from around 2,000 in 2009 to almost 10,000 in 2016. The number of individual articles has in the same period grown from an estimated 8,000 in 2009 to 45,000 in 2016. The growth in article numbers has clearly increased since 2014, after some major research funders in Europe started to introduce new centralized payment schemes for the article processing charges (APCs).

## INTRODUCTION

Ever since the emergence of the world wide web, a quarter of a century ago, academics and scholarly publishers have experimented with different forms of Open Access (OA), a radically new business model for the dissemination of research publications. The common characteristic of all different forms of OA is that the basic building brick of modern research reporting, the peer reviewed article, is available to anybody with Internet access free of charge and access barriers. The different variations of OA include immediate and delayed access, access just to read versus access with extensive reuse rights. OA can be provided with no charges to the authors and their institutions, or publishing charges can be levied to fund the publication. The abbreviation for such charges is APC, which can be interpreted as article processing or article publication charge. In addition to the articles themselves being made OA by the publishers (gold OA), it is also possible that the authors legally upload manuscript copies to institutional or subject-specific repositories (green OA). And as with many other types of web content, illegal copies can be found for many articles behind subscription paywalls

OA content can thus be found through channels which are clear alternatives to subscription journals, or as substitutes for those in the form of manuscript version freely available. But there is one variation which big commercial and society publishers

Corresponding author
Bo-Christer Björk, Bo-Christer.Bjork@hanken.fi

have eagerly embraced in the last few years: the hybrid OA journal. A hybrid journal is a traditional journal, for which readers need a subscription (or where readers can pay to view individual articles). This essentially means that the revenue for the publisher comes mainly from readers or the organizations they work for. But in addition, the journal offers authors the possibility to open up their individual article as OA immediately upon publication, on condition of the payment of a surcharge similar to the APCs paid in many gold OA journals. The vast majority of subscription journals from the leading scholarly publishers are nowadays hybrid. Due to the price level of typically around 3,000 USD, which many authors and their institutions perceive as high (*Tenopir et al., 2017*), the actual uptake of the option has however so far remained low, on average only a couple of percentages of all articles in the journals in question. There are, however, signs that the uptake is on the rise.

The first documented hybrid journals were published by the Entomological Society of America in the late 1990s (*Walker, 1998*). The APCs were low by today's standards, a couple of hundred dollars. David Prosser wrote an interesting analysis in 2003, where he outlined hybrid journals as a risk free transition path towards full OA (*Prosser, 2003*). Then, in a bold move in 2004, Springer announced the hybrid option "Open Choice" for their full portfolio of over 1,000 subscription journals (*Springer, 2004*). Springer was also the first publisher to experiment with deals bundling together subscription e-licenses for whole universities or consortia, with "free" hybrid OA for articles authored by scholars from these universities (*Albandes, 2009*).

Since around 2010 most big publishers have announced similar schemes and the number of journal offering the hybrid possibility has dramatically increased. Starting a hybrid program for established journals is very inexpensive on the journal level, since making the necessary changes to the web submission and publishing systems can be spread over hundreds of journals sharing the same infrastructure. It is also essentially risk-free, in contrast to starting new full OA journals or converting journals, since the subscription revenue remains. Many academics and research funders have been quite critical of academics paying twice for the same content, a practice pejoratively named "double-dipping" (*Cressey, 2009*). Publishers have tried to reassure critics that this is not the case, and that they will decrease subscription prices in proportion to the hybrid uptake (*Elsevier, 2017*). But monitoring this has been very difficult due to the lack of data on the uptake. Publishers have different ways of "tagging" hybrid articles, and counting them manually in almost 10,000 journals is a daunting task. In addition, it would be very useful to see if the hybrid strategy shows any signs of becoming the road towards full conversion to OA that David Prosser had outlined in 2003 (*Prosser, 2003*).

The aim of this study was consequently to study the global number of journals offering the hybrid option and in particular the total number of articles published in them using the option. Due to the fact that the author of this article has carried out two earlier studies of a similar nature, a further aim was to provide reliable estimates of the longitudinal development of both the number of journals and articles since 2009.

**Table 1** Earlier studies incorporated in the results.

| | No. of journals (when studied) | No. of articles (when studied) | No of publishers studied |
|---|---|---|---|
| Dallmeier-Tiessen et al. (2010) | Oct. 2009 | 2009 | 12 |
| Björk (2012) | Febr. 2012 | 2009, 2011 | 15 |
| Laakso & Björk (2016) | Not directly studied | 2011–2013 | 5 |
| RIN (2015) | 2012, 2014 | 2012, 2014 | 32 |

## METHODS

The two aggregate parameters for understanding the evolution of over time of the hybrid OA market, are the number of journals offering the hybrid option and the total number of hybrid articles published yearly. Additional interesting parameters include the share of eligible articles for which the the hybrid option has been used.

It is very difficult to study the number of articles published in hybrid OA journals. Firstly, there is no general index of hybrid journals, so even identifying which journals are hybrid is challenging. The leading commercial publishers usually offer this option for the journals they own themselves, but the situation is less clear for journals that they publish on behalf of scholarly societies. Secondly, there is no standardized way in which hybrid articles are labelled. Publishers indicate hybrid articles in varying ways in tables of content, and a few have search functions across their journal portfolios, which allow selectively picking out hybrid articles. Manually browsing the tables of content of around 10,000 hybrid journals in order to find the hybrid articles for even a sample of the more than one million "eligible" articles published each year is just not feasible.

One option is to find the total numbers from published reports. Many publishers have announced the number of journals on their websites, and occasionally they have announced how many hybrid articles they published in a given year, for instance in press releases. Another option is to count articles using some automated means from the journal websites.

In this study the evolution from 2009 until 2016 is studied, using data from a number of published studies as well as new data collected for this study. The earlier studies are listed in Table 1.

The Study of Open Access Publishing (SOAP) was carried out with EU funding by a consortium of partners which also included publishers (Dallmeier-Tiessen et al., 2010). The scope of the study was open access broadly, but the results also included an estimate of the number of hybrid journals and articles, based partly on data obtained directly from publishers. My own study of the evolution of hybrid OA from 2009 to 2011 followed in the footsteps of the SOAP study, and also used a mixed method of searching publishers' websites and asking them directly for numbers (Björk, 2012). A later study that I carried out together with Mikael Laakso focused more on finding out the uptake level in individual hybrid journals, but it also produced overall article figures for 2011–2013 (Laakso & Björk, 2016). The method differed from the other studies, since it was based on automated identification of hybrid articles, using for instance the labelling of such articles with text

**Table 2** The hybrid publishers included in the study.

| | |
|---|---|
| **5 biggest:** | Elsevier |
| | Springer Nature Group |
| | Wiley-Blackwell |
| | Taylor & Francis |
| | Sage |
| **15 other:** | Emerald |
| | Oxford University Press |
| | Cambridge University Press |
| | BMJ Group |
| | Royal Society |
| | Royal Society of Chemistry |
| | American Physical Society |
| | American Institute of Physics |
| | American Chemical Society |
| | Proceeding of the National Academy of Sciences (PNAS) |
| | International Union of Crystallography |
| | Institute of Physics (IOP) |
| | Company of Biologists |
| | Portland Press |
| | Brill |

like "Creating commons" or "copyright the authors". The RIN study focused on the OA situation for UK authors, but also produced comparative global figures (*RIN, 2015*).

For the present study 15 leading publishers were asked to provide the hybrid article numbers for 2014–2016. In order to increase the chance of getting answers, we promised only to report the total figures, not the figures for individual publishers. For this reason, the results are presented on the aggregate level only.

Twelve of the publishers provided the article numbers. For two of the publishers it was impossible to find out an appropriate contact person, and one publisher refused to provide the data. For these three the figures were estimated from their websites using browsing and/or using the search facilities of the publishing platform in question. In a follow-up query seven of the fifteen publishers also provided the current number of hybrid journals. For the rest it was easier to obtain the journal numbers directly from their websites.

In addition to these, five further publishers, from which the Open Access Scholarly Publishers Association had independently obtained data, were included in the study. The names of all 20 publishers are listed below in Table 2. PNAS is a single journal published by the US National Academy of Sciences, rather than a publisher, but it has been included in the list, since almost a third of the articles are hybrid, implying almost 1,000 hybrid articles per year.

The article numbers of 2009 and 2011 were taken from (*Björk, 2012*) and those of 2014–2016 were collected in this study.

For some of years studied (2010, 2012, 2013), there were no estimates of the total article numbers of all twenty publishers. However, previous studies have provided estimates for

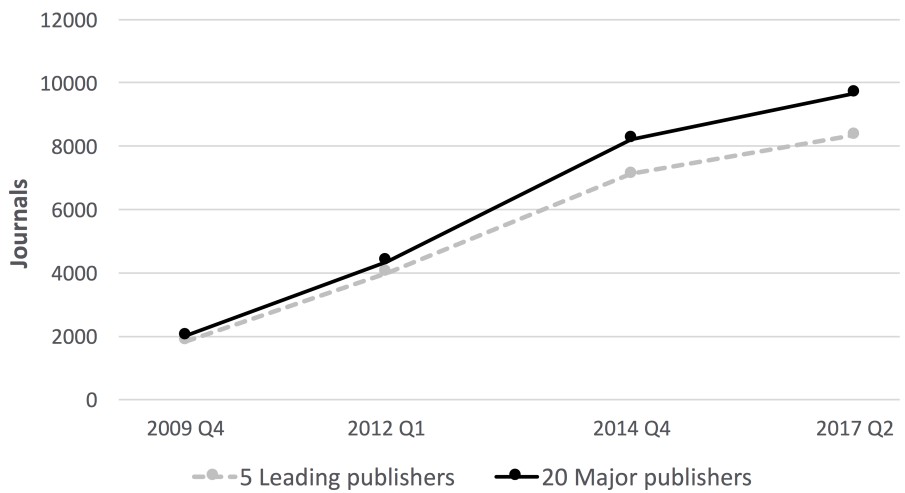

**Figure 1** The evolution of the number of hybrid journals 2009–2017.

the years in question for the five biggest publishers. Furthermore, the ratio of the number of hybrid articles from the big five to all twenty publishers could be calculated from the available estimates for both for 2009, 2011 and 2014–2016. That share has over the years varied between 68 and 80%. Estimates for all twenty for the remaining years could then be extrapolated using approximated longitudinal evolution of the ratio as a basis.

It is not possible to retrospectively estimate the growth in the number of journals on a year to year basis. But the existing earlier estimates can be combined with the new data collected be combined to provide an approximate timeline. The SOAP study (*Dallmeier-Tiessen et al., 2010*) and the earlier study by this author (*Björk, 2012*) provide figures for October 2009 and February 2012. The RIN study (*2015*) provides a figure for approximately the end of 2014. The data for this study was collected in June 2017.

There are many smaller publishers who also have hybrid journals, so the results we report must be understood as lower boundaries. For instance, the RIN study (*2015*) estimates that 49% of all Scopus-indexed journals were hybrid in 2014, which would indicate around 11,000 journals. This can be compared to their estimate for the leading 32 publishers in whose journals UK authors published, which was 8,230. Due to the difficulty in data collection, and since it would be difficult anyway to obtain historical data for the early years, no attempt was made in this study to go beyond the 20 publishers listed.

## RESULTS

The evolution of the number of journals offering the hybrid option is shown in Fig. 1. The data is only available for four discrete points, indicated in the graph.

The figure shows a period of rapid growth between 2009 and 2014, after which the trend has slowed down, probably because most big publishers have already converted the majority of their subscription journals to hybrid journals, reaching a saturation point. The estimate of this study is that twenty leading publishers had 9,678 hybrid journals in June

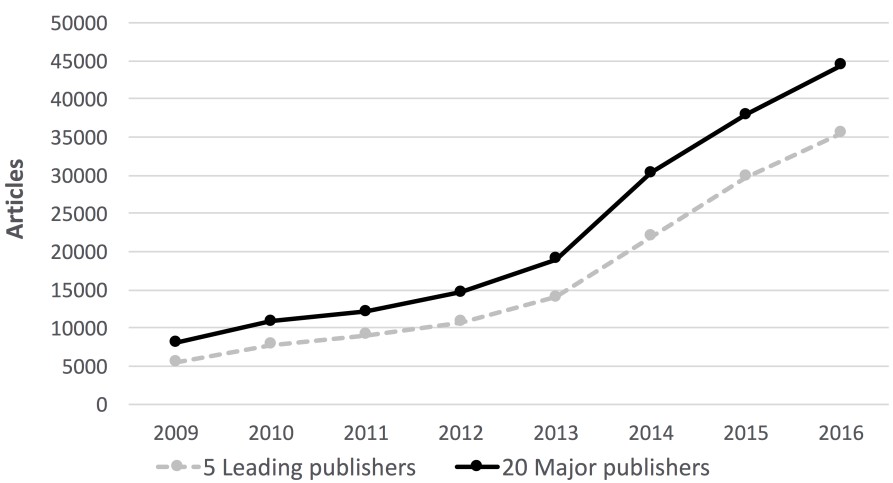

**Figure 2**  The evolution of the number of hybrid articles 2009–2016.

2017. Data in Table 2 of the RIN study (*RIN, 2015*, p. 17) can be used to calculate that, at the end of 2014, 73% of the journals of the big five publishers were hybrid.

The evolution of the number of articles published in these journals (Fig. 2) also shows a sustained growth, but a slightly different pattern.

Compared to the number of journals, the growth rate of articles was slower in the period 2009–2013. But from 2014 onwards, the growth has been faster, meaning that the average number of hybrid articles per journal offering the option has started to grow. By 2016 the total number of articles had reached an estimated 44,395.

All in all the scholarly publishing market has seen an increasing market concentration with the five leading publishers in 2013 accounting for nearly 50% of Web of Science indexed articles (*Larivière, Haustein & Mongeon, 2015*).  Their market share of hybrid articles is even more pronounced.

## DISCUSSION

As long as the authors of hybrid articles have had to secure the funding of the APCs from either personal funds, departmental funds, or external research grants, the uptake levels have remained very low for the vast majority of journals. The rather uniform price level of around 3,000 USD for the extra feature of open access to an article which in any case will be published, seems to most authors like an expensive luxury, unless the journal in question is a very prestigious one. There are many journals with zero articles per year, and the average annual hybrid article share for journals with at least one hybrid article per year was during the period 2011–2013 only 3.8% (*Laakso & Björk, 2016*). But there are interesting cases of journals with a high uptake which have been able to use hybrid as a stepping stone to converting into full OA journals. For instance, Nature Communications was launched as a hybrid and electronic only journal in 2010 and converted to full OA from the start of 2016.

The accelerating growth in hybrid article numbers in 2014–2016 is in addition to the increased number of journals, probably mainly due to two major changes in the funding

infrastructure. Firstly, several leading research funders have set up centralized funds for paying APCs and secondly, a new type of consortial electronic subscription licenses which also include the payment of hybrid APCs has stated to emerge.

The centralized funding scheme makes a break with the earlier practice of making APCs (both for full OA and for hybrid journals) allowable costs in externally funded research grants. That idea has not worked well, because it essentially means that if a researcher does use grant funds for APCs, that correspondingly decreases uses for other items, such as a research assistants or conference trips. The new scheme was pioneered by Wellcome Trust and received considerable UK governmental support following the Finch report (*Finch, 2012*). In the budgets of research funding organizations like the UK research councils and the Austrian Science Fund (FWF), the money is separately budgeted for just APCs on a central level, and payment is more or less automatic. An additional advantage is that the transaction costs of handling APC payments is usually much lower, than in the allowable cost option.

After the introduction of such mechanisms several research funders have started to publish breakdowns of their APC expenditures. For instance Jisc, FWF and Wellcome Trust together funded 5,612 hybrid articles in 2014 and 5,912 articles in 2015 (*Jahn & Tullney, 2016*). Some of these funders have, however, been alarmed by the fact that they are paying more APCs for hybrid than full OA journals, and at higher prices. For this reason, they commissioned a report in 2014 outlining different ways for capping APC levels, in particular for hybrid journals (*Björk & Solomon, 2014*).

The other option, an extension to the subscription e-license to also include APCs for OA articles in hybrid journals, which had been piloted already around 2008–2009 by Springer, is now again becoming popular. Many national library consortia are very keen to negotiate such deals, which also an increasing number of major publishers seem willing to make. The boost to hybrid uptake is demonstrated by for instance the Springer "Compact" deal with a consortium representing 32 Swedish universities (*Neidenmark, 2016*). In 2015 only 157 articles with Swedish corresponding authors were published in Springer hybrid journals. But in 2016 after the new deal was in place, 236 articles in Springer subscription journals automatically became OA already in the first three months of the year. That would mean an estimated almost 1,000 articles per annum.

Due to institutional arrangements such as these the uptake of hybrid OA has started to vary a lot between countries. Some European countries with centralized research funding and strong OA strategies, seem to be leading the development. For instance, UK authors in 2014 published 6.4% of their Scopus-indexed articles as hybrid OA, when the corresponding global average was 2.4% (*RIN, 2015*).

## CONCLUSIONS

Hybrid OA publishing is just a part of the overall changing OA picture. Another important type of gold OA publishing consists of the OA megajournals, which have rapidly emerged in the wake of the phenomenal success of PLOS ONE (*Spezi et al., 2017*). Starting megajournals is for major established publishers another strategic way of

gradually increasing their OA presence, a way in which they can leverage their prestige and infrastructure. Although the number of such journals is low, their combined article volumes and growth rate is quite similar to hybrid OA. Our ongoing research shows that the article volume of 16 leading megajournals has increased from 6,913 in 2010 to 73,703 in 2016 (extended from *Björk, 2015*). Together hybrid OA and megajournal articles constituted around 5.5% of all scholarly articles indexed in Scopus in 2016. The share of articles in Scopus made OA immediately by the publisher is currently around 20%, and a large part of the growth in the past five years has come via the two above categories (our on-going research).

## ACKNOWLEDGEMENTS

This study would have been difficult to carry out without the co-operation of the twelve publishers, which were kind enough to provide the hybrid article counts for 2014–2016. Very useful additional data was provided by Claire Redhead, from the Open Access Scholarly Publisher's Association.

### Funding

The authors received no funding for this work.

### Competing Interests

The authors declare there are no competing interests.

### Author Contributions

- Bo-Christer Björk conceived and designed the experiments, performed the experiments, analyzed the data, contributed reagents/materials/analysis tools, wrote the paper, prepared figures and/or tables, reviewed drafts of the paper, did the whole study alone.

### Data Availability

The raw data has been uploaded as a Supplementary File.

### Supplemental Information

Supplemental information for this article can be found online at http://dx.doi.org/10.7717/peerj.3878#supplemental-information.

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
