# Peer review of "Growth of hybrid open access, 2009–2016"

_PeerJ, doi:10.7717/peerj.3878_

## Round 0.1 · original submission · Minor Revisions

This work addresses the important phenomenon of "hybrid" open access publishing, which as become increasingly popular. All three reviewers provided constructive feedback on the manuscript and several suggestions for revisions. Please engage with their comments and try to address them as best you can.

In addition to the detailed comments of reviewers 1 and 2, the manuscript would be strengthened if the data for individual publishers could be included. It's understandable that this was not done to increase the response rate to the survey, but if it's at all possible to negotiate for at least a subset of these data to be released, I believe it would be of great interest to the community. Also, I would like to reiterate one of the reviewer's suggestion to include some high-level discussion in the paper of what the consequences of the growing hybrid model might be (e.g., how might it change the publishing costs for authors, universities, or the public, and so on).

Thank you for your work on this interesting topic.

Yarden

·

Basic reporting

This is a short report of a study that has attempted to estimate the increase in the numbers of hybrid OA journals and articles in the period 2009-2016. It is a fairly straight-foward account but in a number of places (highlighted with comments in the annotated PDF), the mode of expression could be clearer. In its present form, the manuscript is not an easy read. Greater clarity and attention to sentence structure will make it more accessible for an international audience.

Experimental design

The methods used to estimate numbers of hybrid OA journals and articles include a mix of requests to publishers, estimates derived from examination of publisher web-sites and information gathered from other similar studies. In some places the description of the methods could be clarified for the benefit of the reader (see annotated PDF for details).

The study aggregates estimates from 20 leading publishers (including the big five). It would be useful to know what is their total market share (in publishing and in OA).This would provide a useful sense of scale that cannot be derived from the numbers reported in Figures 1 and 2.

Validity of the findings

The estimates produce appear reasonable to me. However, what is lacking – and this is highlighted by the mention of OA mega-journals in the final paragraph of the discussion – is a fuller sense of context. What fraction of all OA articles (in repositories, in hybrid OA journals and in OA journals) are hybrid? This seems an important and relevant point, especially if one wants to understand the importance and the potential future of hybrid options (and given the concerns expressed by some national funders).

Overall also, I would like to see more references cited in support of some of the claims made (e.g. that authors perceive an APC of $3000 as 'high'; that starting hybrid options involves 'very low cost'; that hybrid APCs prices are 'rather uniform' - see annotated PDF). Otherwise, the tone risks becoming overly rhetorical.

Additional comments

No further comments.

Reviewer 2 ·

Basic reporting

(1) There is a problem with table two. The column headings do not match the data in the table for the the first two columns. This must be fixed.

(2) The author spends a lot of time in the methods discussing the challenges to collecting the full cadre of data that would be ideal for a study like this, and does an appropriate job conveying that because of the limitations the results are a lower bound. However, it is somewhat confusing to make sense of this in the method and I think that to address this the authors should include a THIRD table that lays out the methods used for each year of the study (and of course the citations where appropriate). This information is present in the text but I think it would be more clearly conveyed as a table that is organized by year.

Grammar / Clarity -
- In the abstract the abbreviation OA is used but introduced. This can be done in line 10.
- In line 14 of the abstract what exactly “scattered” is referring to is unclear. Is it that the small numbers are scattered across many journals? This should be stated explicitly.
- Line 28 the “one” should be removed
- The sentence that starts on line 31 is awkward. The point of laying out the different types of OA is important and should be stated clearly. The sentence could be split into two sentences, the first that addresses the “timing” and the second that address the “read vs. re-use rights”.
- Comma after “but in addition” in line 45
- The sentence that starts on line 49 should be rewritten to “ Due to the price level of typically around $3,000 USD, which many authors and their institutions perceive as high, the uptake of the option is low, on average only a couple of percentages of eligible articles”
- Should Entemological Society of America on line 53 have its first letters uppercased?
- Line 59, “starting” should be removed
- Line 62, “Since” instead of “Starting from”
- Line 64, replace “very low cost” with inexpensive
- line 65, consider changing systems to platform
- line 67/68 - end sentence with “since the subscription revenue remains”
- line 90 - almost always or usually instead of “tend in particular”
- line 92 - Secondly,
- line 143 - it is unclear what is being refered to with “That share” on line 143. Is it what fraction of the number of hybrid-OA articles come from the major 5 publishers?
- line 189 - English should be adjusted to “from either personal funds, departmental funds, or external research grants”
- Sentence starting on line 194 - not clear whether the 3.8% average share refers to the percentage of total articles that are OA in a journal over a year or the fraction of journals that have atleast 1 OA article per year
- Sentence starting on line 203 — Not entirely sure if this meaning is correct but if not the sentences can be adjusted. “First, several leading research funders set up centralized funds to pay for APCs. Second, a new type of consortial electronic subscription license has been introduced in which APC costs are included in the price.”

Experimental design

(1) It is necessary to report the total number of article published by the journals along with the number of hybrid-OA articles. Without the total number of articles published we cannot distinguish between whether the rate of uptake has increased or whether the rate of uptake has remained the same but the total number of articles published has increased.

(2) The author mentions that PNAS is a unique publication in that close to 1/3 of its articles are hybrid open-access. This raises the concern that perhaps it is such an outlier compared to the other publishers/journal groups that is skewing the overall statistics.

Validity of the findings

The paper “Growth of hybrid open access, 2009-2016” provides an overview of the development of the hybrid open access sector of publishing and presents new data about both the number of journals that offer hybrid open access and the number of hybrid open access articles. The author does a good job conveying the challenges of acquiring the necessary data about the hybrid-OA sector. And given the clear challenges to acquiring this data, it is important that this work be published so that these numbers are available to the public.

My concerns related to this section covered in the sections above.

Additional comments

No comment

·

Basic reporting

No comment, in general, this paper presents solid information regarding the Hybrid Open Access model that might be relevant to many scholars focusing on academic publishing.

Experimental design

This is a descriptive manuscript with interesting statistics which were obtained using data that is not public which makes it hard to replicate. I would negotiate with the publishers to also allow access to the underlying data.

Validity of the findings

Besides the fact the there is a growing number of publishers using the Hybrid Model which, in itself, is relevant, it lacks substance as to what the impact this phenomenon has in the publishing world as a whole.

Additional comments

It is a nice manuscript but a little more argumentation would make it even better.

---

## Round 0.2 · accepted · Accept

While in production, please consider Reviewer 1's additional comments on your revised manuscript.

Also, please clearly state in the Abstract as well as in the Introduction that the "APC" acronym stands for "Article Processing Charges".

·

Basic reporting

Since this is a review of a revised manuscript I will confine my comments to a single section. Overall, I am satisfied that the author has addressed all the comments made on the first version of this article and I think it is now suitable for publication. There are one or two residual points – all minor – which I leave to the author's discretion. (For detailed see the comments in the attached manuscript file).

Experimental design

Please see above.

Validity of the findings

Please see above.